# SCALING LAWS FOR PREDICTING DOWNSTREAM PERFORMANCE IN LLMS

## ABSTRACT

Precise estimation of downstream performance in large language models (LLMs) prior to training is essential for guiding their development process. Scaling laws analysis utilizes the statistics of a series of significantly smaller sampling language models (LMs) to predict the performance of the target LLM. For downstream performance prediction, the critical challenge lies in the emergent abilities in LLMs that occur beyond task-specific computational thresholds. In this work, we focus on the pre-training loss as a more computation-efficient metric for performance estimation. Our two-stage approach consists of first estimating a function that maps computational resources (*e.g.,* **F**LOPs) to the pre-training **L**oss using a series of sampling models, followed by mapping the pre-training loss to downstream task **P**erformance after the critical "emergent phase". In preliminary experiments, this **FLP** solution accurately predicts the performance of LLMs with 7B and 13B parameters using a series of sampling LMs up to 3B, achieving error margins of 5% and 10%, respectively, and significantly outperforming the FLOPs-to-Performance approach. This motivates **FLP-M**, a fundamental approach for performance prediction that addresses the practical need to integrate datasets from multiple sources during pre-training, specifically blending general corpora with code data to accurately represent the common necessity. `FLP-M` extends the power law analytical function to predict domain-specific pre-training loss based on FLOPs across data sources, and employs a two-layer neural network to model the non-linear relationship between multiple domain-specific loss and downstream performance. By utilizing a 3B LLM trained on a specific ratio and a series of smaller sampling LMs, `FLP-M` can effectively forecast the performance of 3B and 7B LLMs across various data mixtures for most benchmarks within 10% error margins.

## 1 INTRODUCTION

Large language models (LLMs) form the basis for numerous real-world applications (Brown et al., 2020; Jiang et al., 2023; Touvron et al., 2023) and scaling laws analysis serves as the foundation for LLMs development (Kaplan et al., 2020; Bahri et al., 2024). The key idea of scaling laws involves training a sequence of language models (LMs) to gather data (*e.g.,* expended compute and corresponding model performance). This data is then used to build a predictive model that estimates the performance of a substantially larger target LLM (Su et al., 2024; Hoffmann et al., 2022).

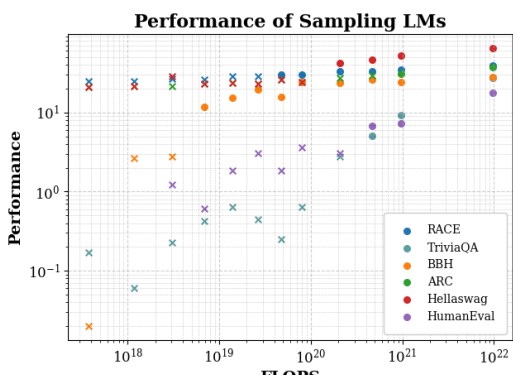

Figure 1: The performance of sampling LMs with increasing compute. x represents non-emerged data points, and • indicates emerged data points that surpass a randomness threshold of 5.

Previous efforts focus on predicting the target LLM's pre-training loss and establish a power-law relation between the computational resource expended (*e.g.,* floating-point operations per second (FLOPs)) and the final loss achieved (Kaplan et al., 2020; Muennighoff et al., 2024; Henighan et al., 2020). Further, we aim to predict the downstream performance in LLMs to more accurately reflect the primary concerns regarding their

capabilities. The critical challenge is the emergent abilities in LLMs, which states that LLMs only exceed random performance when the FLOPs expended during training surpass task-specific thresholds (Wei et al., 2022). Supposing a task threshold of $F_c$, typical methods require training $N$ LMs, expending total FLOPs $F_t = \sum_{i=1}^{N} \text{FLOPs}_i > N \times F_c$, to obtain $N$ effective data points, thereby necessitating significant computational resources. Fig. 1 demonstrates that the sampling LMs require more than $5 \times 10^{20}$ FLOPs to perform better than random on most benchmarks, with only three data points available to fit the predictive curve across these benchmarks. Hu et al. (2023) address this challenge by significantly increasing the sampling times to compute the `PassUntil` of a task, basically increasing the "metric resolution" to enable the abilities to emerge earlier (*i.e.,* reducing $F_c$). However, this approach faces challenges in translating the `PassUntil` back to the original task metric of concerns and requires huge amounts of FLOPs spent on sampling.

In this work, we target the actual task performance prediction based on two intuitions: (1) Predicting the target pre-training loss is easier and achievable since there is no "emergent phase" in the pre-training loss, as extensively verified in Kaplan et al. (2020); Hoffmann et al. (2022); (2) There is an observed correlation between the pre-training loss and the downstream task performance after the "emergent point" (*i.e.,* the pre-training loss goes below a critical threshold) (Du et al., 2024; Huang et al., 2024). Rethinking previous practice, training LM $i$ to convergence requires expending FLOPs$_i$ for obtaining a single data point. In contrast, our approach stems from a crucial insight: **Collecting (pre-training loss, performance) data points at intermediate checkpoints prevents the need for fully training LMs to convergence, thereby enhancing sample efficiency.** Essentially, we can collect a huge amount of useful data between the initial point of above-random performance in LMs and their convergence point for performance prediction.

Thus, our approach consists of two sequential stages: (1) **FLOPs → Loss**: Predict the target pre-training loss based on the expended FLOPs. Following previous work, we train a series of sampling LMs within the same model family to develop a power-law predictive model. For this stage, the expended FLOPs are not required to reach above the emergent threshold. (2) **Loss → Performance**: Predict the downstream performance based on the pre-training loss. We collect data points from intermediate checkpoints of various sampling LMs that exhibit above-random performance, and develop a regression model for prediction. In preliminary experiments with sampling LMs up to 3B, this **FLP** solution predicts the performance of 7B and 13B LLMs across various benchmarks with error margins of 5% and 10% respectively, significantly outperforming direct FLOPs-to-Performance predictions.

Motivated by these findings, we present **FLP-M**, a fundamental solution for performance prediction that addresses the growing demand for integrating diverse datasets during LLMs pre-training, focusing on integrating the general corpus with code data in this work. FLP-M targets fine-grained domain-specific pre-training loss to capture the performance changes. Specifically, we extend the power law analytical function to predict the domain-specific loss based on FLOPs across multiple data sources. Then we employ a two-layer neural network to model the non-linear relationship between multiple domain-specific loss and the downstream performance. Through evaluation, we demonstrate that FLP-M effectively predicts the performance of 3B and 7B LLMs trained on various data mixtures (within 10% error margins for most benchmarks). This is achieved by utilizing a 3B LLM trained on a specific data mixing ratio along with a series of smaller sampling LMs.

## 2 RELATED WORK

### 2.1 SCALING LAWS

Estimating the performance of the target LLM prior to training is essential due to the significant resources required for pre-training (Minaee et al., 2024; Wan et al., 2023). The scaling laws of LLMs guide the systematic exploration in scaling up computational resources, data, and model sizes (Kaplan et al., 2020; Hestness et al., 2017). Previous efforts in this filed demonstrate that LLMs' final pre-training loss on a held-out validation set decreases with an increase in expended FLOPs during pre-training (Kaplan et al., 2020; Hoffmann et al., 2022; Yao et al., 2023). The following work subsequently establishes the scaling laws for computer vision models (Zhai et al., 2022), vision-language models (Henighan et al., 2020; Alabdulmohsin et al., 2022; Li et al., 2024a), graph self-supervised learning (Ma et al., 2024), reward modeling (Gao et al., 2023a; Rafailov et al., 2024), data filtering (Goyal et al., 2024), knowledge capabilities of LLMs (Allen-Zhu

& Li, 2024), data-constrained LMs (Muennighoff et al., 2024), data poisoning (Bowen et al., 2024), LLMs vocabulary size (Tao et al., 2024), retrieval-augmented LLMs (Shao et al., 2024), continued pre-training of LLMs (Que et al., 2024), LLMs training steps (Tissue et al., 2024), fine-tuning LLMs (Tay et al., 2021; Lin et al., 2024; Hernandez et al., 2021), learning from repeated data (Hernandez et al., 2022), the sparse auto-encoders (Gao et al., 2024), hyper-parameters in LLMs pre-training (Yang et al., 2022; Lingle, 2024), and the mixture-of-expert LLMs (Clark et al., 2022; Frantar et al., 2023; Yun et al., 2024; Krajewski et al., 2024).

Despite the efforts, directly estimating the downstream performance of LLMs more accurately reflects the models' capabilities pertinent to our concerns, yet it confronts challenges associated with emergent abilities in LLMs (Wei et al., 2022). In general, the compute required for pre-training must surpass a task-specific threshold to enable pre-trained LMs to perform better than random chance. Previous work addresses this challenge by using the answer loss as an alternative metric (Schaeffer et al., 2024) or increasing the metric resolution, such as measuring the average number of attempts to solve the task (Hu et al., 2023). However, they encounter difficulties in aligning the proposed metric with the original task metric, which is of paramount interest to us. Our research directly predicts the task performance metrics of the target LLMs by utilizing readily available intermediate LMs. This approach operates independently from and complements existing approaches.

## 2.2 DATA MIXTURE

Creating the pre-training dataset necessities collecting data from different sources (Liu et al., 2023; Shen et al., 2023; Bi et al., 2024; Wei et al., 2023), making the data mixture a critical factor in the study of scaling laws. Ye et al. (2024) propose the data mixing laws to predict the pre-training loss of the target LLM given the mixing ratios. Liu et al. (2024) build the regression model to predict the optimal data mixture regarding the pre-training loss optimization, and Kang et al. (2024) further show that the optimal data composition depends on the scale of compute. In this work, we focus on integrating the data mixture factor to better predict the downstream performance.

## 3 FLP: DOWNSTREAM PERFORMANCE PREDICTION

We introduce a *two-stage* approach to predicting downstream performance in LLMs based on two established findings: (1) Predicting the target pre-training loss and establishing the power-law relation is feasible as it does not involve an emergent phase (Kaplan et al., 2020; Hoffmann et al., 2022). (2) When pre-training loss goes below a task-specific threshold, there is an observed correlation between pre-training loss and downstream task performance (Du et al., 2024; Huang et al., 2024). In this section, we present FLP as a proof-of-concept for this framework with a straightforward implementation.

### 3.1 FLOPs → LOSS

We follow the previous practice to use the analytical power law function to characterize the relation between expended FLOPs $C$ and the pre-training loss $L$:

$$L(C) = \left( \frac{C}{C_N} \right)^{\alpha_N},\tag{1}$$

where $C_N$ and $\alpha_N$ are constant terms to be estimated. In FLP, we train a series of $N$ LMs within the same model family in the same pre-training distribution, progressively increasing model size and training tokens to achieve even sampling. Then we measure their pre-training loss in our curated validation dataset to obtain $N$ pairs of $(C_i, L_i)$ to estimate the constants in Eq. 1.

### 3.2 LOSS → PERFORMANCE

Based on our empirical observation of the scatter plots showing (pre-training loss, performance) data points (see §A), we select the analytical linear function to characterize the relation between the pre-training loss $L$ on general validation data and the task performance $P$:

$$P(L) = w_0 + w_1 * L,\tag{2}$$

where $w_0$ and $w_1$ are constant terms to be estimated. In FLP, we fetch the intermediate checkpoints of each sampling LM, and measure its task performance and pre-training loss. If the performance $P_i$

Table 1: The configurations of the sampling and target LMs with various sizes. HD denotes the hidden dimension, BS denotes the batch size, and LR denotes the learning rate.

| Model Size | #Layer | HD | #Head | FFN | #Tokens | Non-embedding FLOPs | BS | LR |
|---|---|---|---|---|---|---|---|---|
| 43M | 3 | 384 | 3 | 1032 | 8,021,606,400 | 3.70504E+17 | 448 | 0.0052 |
| 64M | 4 | 512 | 4 | 1376 | 11,714,691,072 | 1.18417E+18 | 544 | 0.0042 |
| 89M | 5 | 640 | 5 | 1720 | 16,184,770,560 | 3.03607E+18 | 576 | 0.0038 |
| 0.12B | 6 | 768 | 6 | 2064 | 21,799,895,040 | 6.81931E+18 | 640 | 0.0040 |
| 0.15B | 7 | 896 | 7 | 2408 | 28,846,325,760 | 1.39581E+19 | 672 | 0.0042 |
| 0.2B | 8 | 1024 | 8 | 2752 | 37,213,962,240 | 2.63435E+19 | 736 | 0.0036 |
| 0.25B | 9 | 1152 | 9 | 3096 | 47,563,407,360 | 4.71817E+19 | 768 | 0.0034 |
| 0.32B | 10 | 1280 | 10 | 3440 | 59,674,460,160 | 8.01571E+19 | 800 | 0.0028 |
| 0.5B | 12 | 1536 | 12 | 4128 | 90,502,594,560 | 2.05963E+20 | 960 | 0.0023 |
| 0.72B | 14 | 1792 | 14 | 4816 | 132,026,204,160 | 4.70331E+20 | 1024 | 0.0019 |
| 1B | 16 | 2048 | 16 | 5504 | 185,535,037,440 | 9.75926E+20 | 1152 | 0.0016 |
| 3B | 24 | 3072 | 24 | 8256 | 556,793,856,000 | 9.63212E+21 | 1536 | 0.0004 |
| 7B | 32 | 4096 | 32 | 11008 | 1,258,291,200,000 | 5.09208E+22 | 2048 | 0.0003 |
| 13B | 40 | 5120 | 40 | 13824 | 1,258,291,200,000 | 9.89592E+22 | 2048 | 0.0003 |

of $LM_i$ exceeds the random performance, we can obtain one effective data point $(L_i, P_i)$ to estimate the constants in Eq. 2, where $L_i$ is the pre-training loss of $LM_i$.

## 4 VALIDATION OF FLP FRAMEWORK

### 4.1 SAMPLING AND TARGET LMS

We train a series of 12 sampling LMs up to 3B parameters to predict the performance of target LLMs with 7B and 13B parameters. The configurations of LMs are shown in Tab. 1. We first determine the number of training tokens required for the 7B LLM (approximately 180 times the model size), considering practical needs and inference-time costs. In real-world applications, prioritizing inference efficiency often involves training smaller LMs with a higher token-to-parameter ratio beyond the optimal factor of 20x (Hoffmann et al., 2022). Our preliminary experiments indicate that scaling laws remain applicable even in this over-training regime (within 2.8% error margins). We then proportionally scale down this number to determine the required training tokens for the sampling LMs.

### 4.2 DATA: PRE-TRAINING, VALIDATION, EVALUATION

**Pre-Training** We use the RedPajama v1 (Computer, 2023), which consists of 1.2T tokens in total, and the data is sourced from Arxiv, C4, Common Crawl, GitHub, Stack Exchange, and Wikipedia.

**Validation** We curate a validation dataset to measure the final pre-training loss, which includes 5 distinct domains: math, code, scientific paper, Wikipedia, and general language corpus. Specifically, we utilize subsets from GitHub, ArXiv, Wikipedia, and the English portion of C4, all from the RedPajama validation sets, along with Proof Pile (Touvron et al., 2023) for the math domain.

**Evaluation** We select the following tasks for evaluation, covering fundamental capabilities in LLMs (*e.g.,* knowledge, reasoning, coding): RACE (Lai et al., 2017), TriviaQA (Joshi et al., 2017), BigBench-Challenge (BBH) (Suzgun et al., 2022), ARC-Challenge (ARC) (Clark et al., 2018), Hellaswag (Zellers et al., 2019),

Table 2: The evaluation settings of the benchmarks.

| Dataset | Evaluation Type | Evaluation Method | Metric | Random Performance |
|---|---|---|---|---|
| ARC | Multiple Choice | 10-shot | Accuracy | 25 |
| BBH | Generation | CoT-3-shot | ExactMatch | 0 |
| Hellaswag | Multiple Choice | 10-shot | Accuracy | 25 |
| HumanEval | Generation | 0-shot | Pass@100 | 0 |
| RACE | Multiple Choice | 0-shot | Accuracy | 25 |
| TriviaQA | Generation | 0-shot | ExactMatch | 0 |

and HumanEval (Chen et al., 2021). The evaluation settings for these benchmarks are listed in Tab. 2. We adopt lm-evaluation-harness (Gao et al., 2023b) for unified evaluation.

### 4.3 EXPERIMENTAL SETTING

**Baseline** We consider directly using the expended FLOPs $C$ to predict the downstream performance $P$, and experiment with the following analytical form for comparison:

$$P(C) = (\frac{C}{C_M})^{\alpha_M}, \tag{3}$$

where $C_M$ and $\alpha_M$ are constant terms to be estimated. We denote this approach as **FP**.

**Implementation of FLP** To fit the FLOPs-to-Loss curve, we utilize the final checkpoints from each sampling LM. In addition, during LMs training, a checkpoint is saved at every 1/30th increment

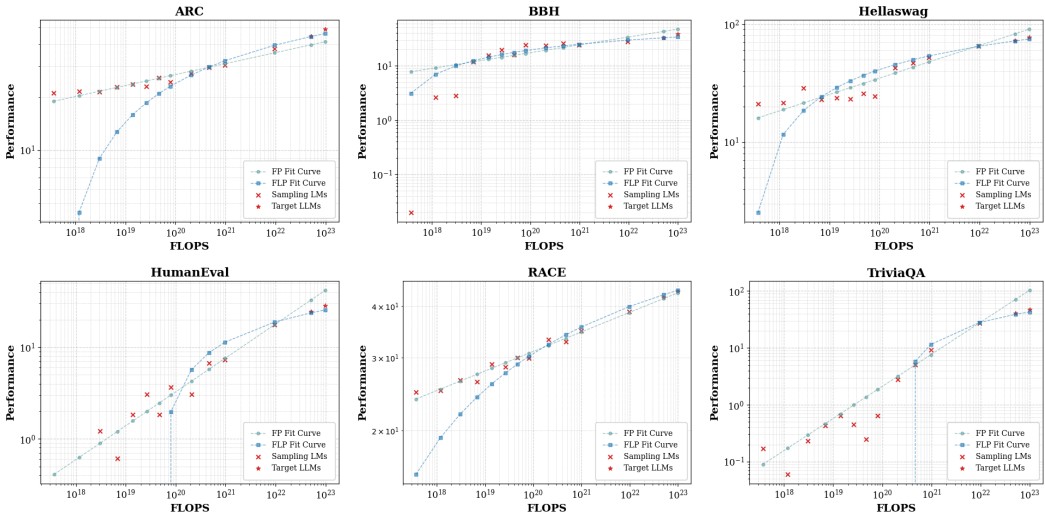

Figure 2: The downstream performance prediction using FP and FLP fit curves. FLP can better predict the downstream performance of target 7B and 13B LLMs across all evaluation benchmarks.

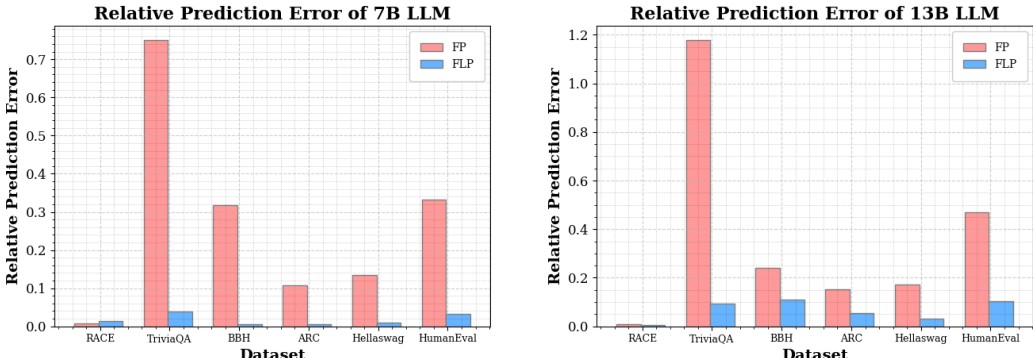

Figure 3: The relative prediction error of 7B and 13B LLMs. FLP achieves a more accurate prediction with error margins of 5% and 10% across all benchmarks for two LLMs respectively.

of the total training progress. We monitor and record the pre-training loss on the training dataset, rounded to two decimal places. Only those checkpoints demonstrating an improvement in pre-training loss are retained. For these selected checkpoints, we evaluate the downstream performance and pre-training loss on the validation set. We then discard those that do not surpass the random benchmark performance by at least 5, and use the remaining data points to fit the Loss-to-Performance curve.

**Evaluation Metrics** In addition to presenting the fitting curves for intuitive visualization. we quantify the prediction accuracy by measuring the relative prediction error:

$$\text{Relative Prediction Error} = \frac{|\text{Predictive Metric} - \text{Actual Metric}|}{\text{Actual Metric}} \tag{4}$$

## 4.4 RESULTS

The downstream performance prediction results are visualized in Fig. 2. Across all evaluation tasks, FLP fit curve can better predict the performance of target LLMs with 7B and 13B parameters using the sampling LMs up to 3B. In contrast, while FP more effectively fits the data points of sampling LMs, it has difficulty accounting for the "emergent phase" characterized by rapid performance shifts, due to the scarcity of data points from this period. As a solution, FLP utilizes pre-training loss as a more fine-grained indicator to monitor performance changes and effectively incorporates data from intermediate checkpoints, enhancing sample efficiency. The evaluation results of relative prediction error are shown in Fig. 3. Unlike the suboptimal predictions of FP, FLP delivers precise forecasts, maintaining relative error margins of 5% and 10% across all benchmarks for 7B and 13B LLMs, respectively.

Compared to FP, `FLP` is less effective at fitting the data points of sampling LMs, especially in HumanEval and TriviaQA. The reason is that we do not align with the "non-emergent" phase of the Loss-to-Performance curve, where LMs exhibit random performance when pre-training loss is beyond the task-specific threshold. Thus, `FLP` predicts higher pre-training loss for LMs with fewer FLOPs, resulting in below-random performance. This issue is not within the scope of `FLP`, as it is specifically designed to predict the performance of LLMs trained with significantly larger FLOPs in practice.

In addition, we discuss additional results in Appendix for the presentation purpose since adding these data points may distort the vertical axis scaling in Fig. 2. We compare `FLP` further with the analytical forms and approaches proposed in GPT-4 (Achiam et al., 2023) and Llama-3 (Dubey et al., 2024) technical reports. The results are shown in §B and §C respectively. We also evaluate the feasibility of employing `FLP` to predict the performance of a 13B LLM on MMLU (Hendrycks et al., 2020), using intermediate checkpoints from a 7B LLM (§D). Overall, the results demonstrate the general effectiveness and applicability of `FLP`.

## 5 `FLP-M`: DATA MIXING FOR DOWNSTREAM PERFORMANCE PREDICTION

Motivated by the encouraging results of `FLP` (§4), we propose `FLP-M`, a fundamental approach to meet the practical needs of integrating data from various sources (Groeneveld et al., 2024; Penedo et al., 2024). In our work, we focus on mixing general corpus with code data, considering two distinct yet overlapping data sources. This intersection offers a more realistic perspective than treating them as distinct domains (Ye et al., 2024), as real-world corpus often spans multiple domains, necessitating an analysis of the interdependence between data sources when formulating our analytical functions.

Compared to the straightforward implementation of `FLP` (§3), `FLP-M` operates on fine-grained, domain-specific pre-training loss, due to the observation that the average loss on the entire validation set fails to effectively reflect performance variations in downstream tasks in the data mixing context (§7.2). This may be due to the fact that changes in pre-training data mixtures simultaneously impact multiple capabilities of the LMs. For instance, an increase in code data loss coupled with a decrease in general data loss may leave the average validation loss unchanged, yet result in LMs with distinct capabilities and downstream performance. Note that unlike the pre-training data mixture, the validation set is deliberately curated by domain, as creating smaller, domain-specific validation sets is manageable.

### 5.1 FLOPs → DOMAIN LOSS

Given the FLOPs $C^G$ spent on the general corpus and $C^C$ spent on the code data, we naturally extend the power law function to the following analytical form to predict the domain-specific pre-training loss $L^D$ on domain $D$:

$$L^D(C^G, C^C) = (\frac{C^G + C^C}{C_T})^{\alpha_C} \times (\frac{C^G}{C_G})^{\alpha_{C_1}} \times (\frac{C^C}{C_C})^{\alpha_{C_2}} \tag{5}$$

where $C_T$, $C_G$, $C_C$, $\alpha_C$, $\alpha_{C_1}$, and $\alpha_{C_2}$ are constants to be estimated. In `FLP-M`, we first select a sequence of total compute $\{C_i\}_{i=1}^N$ spent on pre-training. For each selected $C_i$, we experiment with various ratios to mix two data sources, and decompose $C_i$ into $C_i^G$ and $C_i^C$. We measure the domain-specific pre-training loss $L_i^D$ on a domain-specific subset $D$ of validation data to obtain $(C_i^G, C_i^C, L_i^D)$ data pairs. Then we can estimate the constants in Eq. 5. We also experiment with other potential analytical forms in §7.2.

### 5.2 DOMAIN LOSS → PERFORMANCE

Given the pre-training loss $\{L^D\}_{D=1}^K$ on $K$ domains, we train a two-layer neural network with a hidden layer size of 3 and the ReLU activation function (Agarap, 2018) to predict the downstream performance. The network is optimized using the regression loss with $L_2$ regularization and the Adam optimizer (Diederik, 2014), employing a learning rate of 0.05 that linearly decays to 0 within 2,000 steps and a weight decay of 0.01. In `FLP-M`, we adopt the same strategy as in `FLP` to fetch the intermediate checkpoints and only retain the results that the LMs achieve above-random performance (see §3). Thus, for $LM_i$, we can obtain a sequence of effective data points $(\{L_i^D\}_{D=1}^K, P_i)$, where $L_i^D$ is the pre-training loss on domain $D$ and $P_i$ is the LM's performance. Then we can use these data points to train the neural network. We also explore other functions for fitting in §7.2.

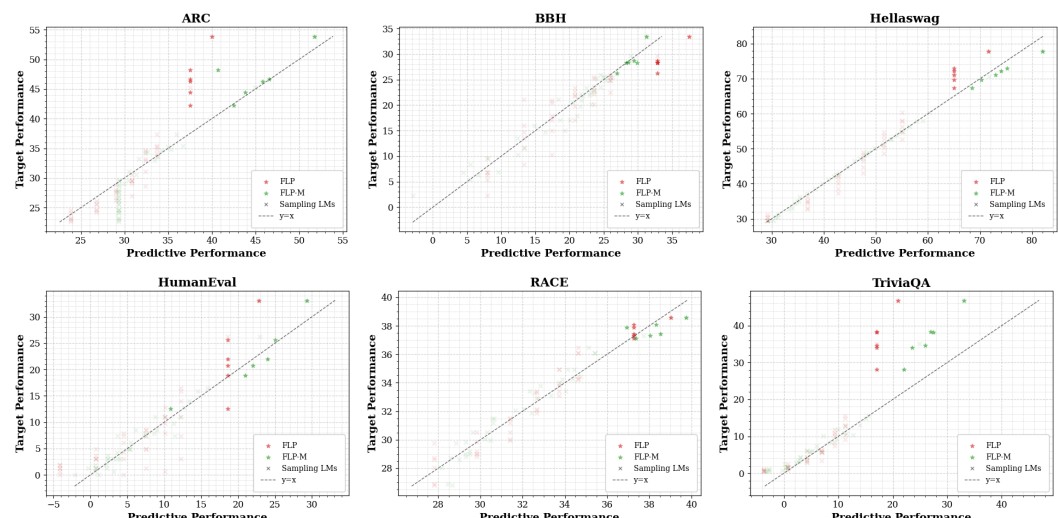

Figure 4: The downstream performance prediction using `FLP` and `FLP-M` fit curves. `FLP-M` can better predict the downstream performance of target LLMs across various data mixing ratios.

## 6 EXPERIMENT FOR `FLP-M`

### 6.1 SAMPLING AND TARGET LMS

We train a series of sampling LMs with sizes of $\{0.12B, 0.2B, 0.32B, 0.5B, 0.72B, 1B\}$, and the corresponding training token numbers are shown in Tab. 1. We train the LMs on the general and code data mixture with $\{0, 0.1, 0.2, 0.3, 0.4, 0.5\}$ as the mixing ratios of code data to reflect real-world usage. We also add one sampling LM of 3B size and 0.3 mixing ratio. For evaluation, we train 3B LLMs with the other mixing ratios and a 7B LLM with 0.3 as the mixing ratio due to the limited compute budget.

### 6.2 DATA: PRE-TRAINING, VALIDATION, EVALUATION

**Pre-Training** For general corpus, we use DCLM (Li et al., 2024b), a curated high-quality pre-training corpus including heuristic cleaning, filtering, deduplication, and model-based filtering. For code data, we use The Stack v2 (Lozhkov et al., 2024), which initially contains over 3B files in 600+ programming and markup languages, created as part of the BigCode project. We mix these two data sources to create the pre-training data mixture using the ratios specified in §6.1.

**Validation** We use the same validation data mixture specified in §4.2 that includes 5 distinct domains.

**Evaluation** The evaluation benchmarks and settings are the same as those in §4.2.

### 6.3 EXPERIMENTAL SETTING

**Baseline** We implement `FLP` within this data mixing context as a baseline, which first predicts the average pre-training loss on the validation set and uses this to estimate downstream performance via linear regression.

**Implementation of `FLP-M`** We adopt the same implementation as in `FLP` (details in §4.3). The distinction is that we individually measure the pre-training loss on each domain of the validation mixture.

### 6.4 RESULTS

The downstream performance prediction results are visualized in Fig. 4. We update the x-axis to "predicted performance" to improve clarity, as the presence of two variables ($C^G$, $C^C$) complicated 3D visualization. Overall, we find that `FLP-M` demonstrates better performance compared to `FLP` when considering the data mixing as an extra factor in scaling laws analysis. Using average validation loss as an indicator for assessing the performance of LMs pre-trained on mixed data sources, such

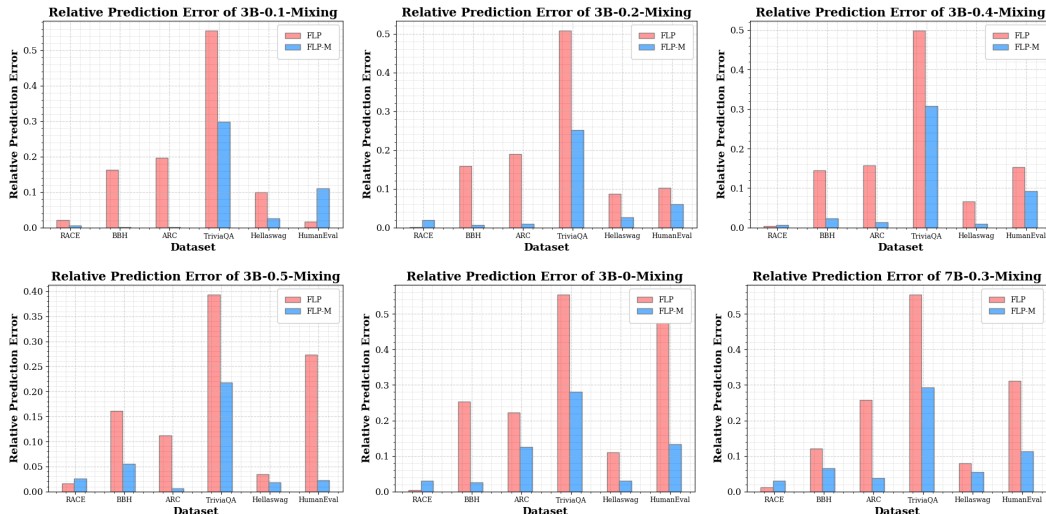

Figure 5: The relative prediction error of downstream performance prediction using `FLP` and `FLP-M`. `FLP-M` can better predict the performance of target LLMs across various data mixing ratios.

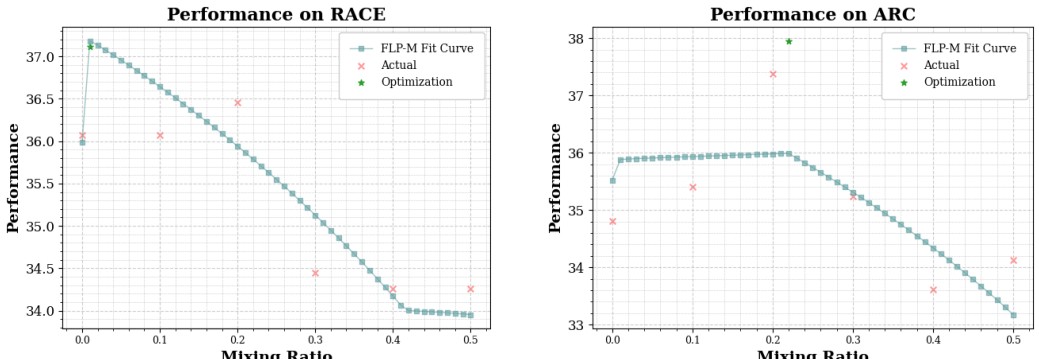

Figure 6: We use the scaling laws function derived via `FLP-M` to find the optimal data mixing ratio that yields the estimated best performance on the corresponding benchmarks.

as general text and code, is limited. Thus, the average loss fails to trace performance variations in downstream tasks because changes in data mixtures can affect different capabilities of the LMs. In contrast, `FLP-M` effectively leverages the domain-specific validation loss to capture the capabilities improvement in LMs, and thus can better predict the downstream performance. In our experiments, `FLP-M` accurately predicts the performance of 3B LLMs across various data mixtures and the 7B LLM with 0.3 data mixing ratio with error margins within 10% for most benchmarks.

However, on TriviaQA, despite significantly outperforming `FLP`, `FLP-M` shows higher relative prediction error, ranging from 20% to 30%. This discrepancy can be explained by the substantial performance improvement when scaling LLMs from under 1B to 3B parameters (increasing from below 12 to over 28). In our sampling LMs configurations (see Tab. 1), we lack sufficient data points to adequately characterize the phase of accelerated performance improvement. To better model this trend, a practical solution is to add several sampling LMs between 1B and 3B parameters.

# 7 FURTHER ANALYSIS

## 7.1 OPTIMIZING DATA MIXTURE USING `FLP-M` SCALING LAWS

We demonstrate how the derived scaling laws using `FLP-M` can be effectively applied to optimize data mixtures, enhancing downstream performance. We focus on 1B LMs in this analysis due to compute constraints. For each dataset, we use the `FLP-M` to estimate the function that maps expended

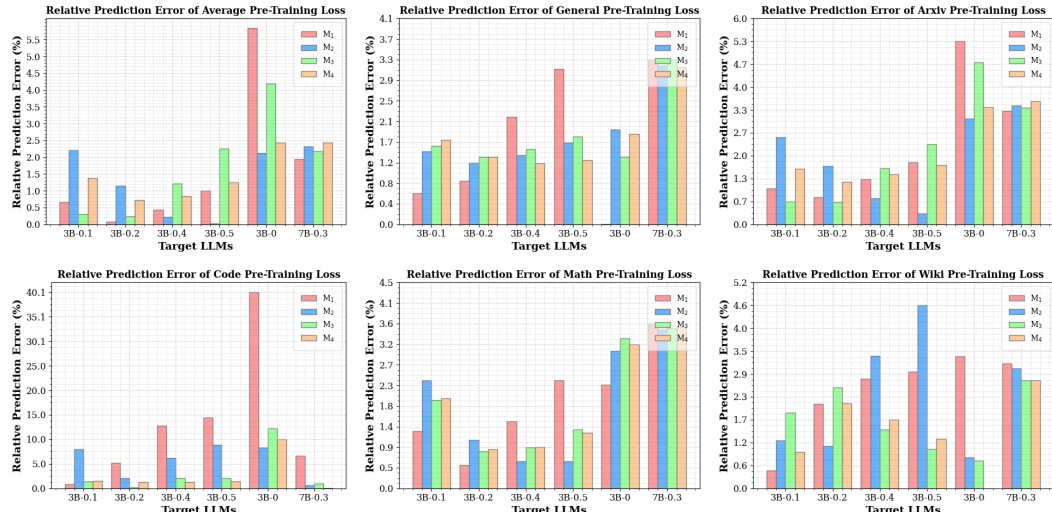

Figure 7: The relative prediction error of average and domain-specific pre-training loss. $M_4$ provides more stable and overall more accurate predictions for domain-specific loss (within 2.5% relative prediction error across most domains).

FLOPs in each data source to the downstream performance. Then we use this function to predict performance across mixing ratios from 0 to 0.5, in intervals of 0.01.

Among all evaluation datasets, the estimated scaling laws function exhibits non-monotonic behavior on the RACE and ARC datasets, reaching its peak at mixing ratios of 0.01 and 0.22, respectively. To verify, we train 1B LMs with these two mixing ratios and measure their performance on the corresponding benchmarks. The results are shown in Fig. 6. We find that the selected optimal mixing ratio can reliably yield better performance compared to the six mixing ratios adopted for the sampling LMs, highlighting FLP-M as a practical approach for optimizing data mixtures to enhance performance on specific target tasks.

## 7.2 ABLATION STUDY

We conduct further analysis to better understand the two stages in FLP-M. Specifically, we compare various approaches to estimate the FLOPs-to-Loss and Loss-to-Performance curves in FLP-M.

**FLP-M: FLOPs → Loss** We experiment with several candidate analytical forms listed in Tab. 3. We assess their performance in estimating the average pre-training loss across the entire validation set, as well as the domain-specific pre-training losses on corresponding subsets. We present the fit curves in Fig. 13 (§E), and the relative prediction errors for pre-training loss estimation are shown in Fig. 7. For average pre-training loss prediction, using more complex analytical models that account for the individual impact of each data source can lead to performance degradation. However,

Table 3: Candidate analytical forms for fitting the FLOPs-to-Loss curve. Except for $C^G$ and $C^C$ representing the compute used for general and code data sources, other constants need to be estimated. The average error is computed across all domains and model types.

| $L^D(C^G, C^C) =$ | Analytical Form | Average Error |
|---|---|---|
| M1 | $(\frac{C^G + C^C}{C_T})^{\alpha_C}$ | 0.029 |
| M2 | $(\frac{C^G}{C_G})^{\alpha_{C1}} \times (\frac{C^C}{C_C})^{\alpha_{C2}}$ | 0.026 |
| M3 | $(\frac{w_0 * C^G + w_1 * C^C}{C_T})^{\alpha_C}$ | 0.017 |
| M4 (Ours) | $(\frac{C^G + C^C}{C_T})^{\alpha_C} \times (\frac{C^G}{C_G})^{\alpha_{C1}} \times (\frac{C^C}{C_C})^{\alpha_{C2}}$ | **0.014** |

relying solely on the total compute for prediction ($M_1$) can cause high prediction errors in certain domains (*e.g.,* code) and are not stable for various mixing ratios. More complex analytical models generally perform better in predicting domain-specific loss. Among them, $M_4$, the adopted model in FLP-M, provides more stable (within 2.5% relative prediction error across most domains) and overall more accurate predictions (achieving the lowest average error shown in Tab. 3).

**FLP-M: Loss → Performance** We experiment with various approaches to estimate the function that maps the pre-training loss to the downstream performance. In this study, we utilize the actual

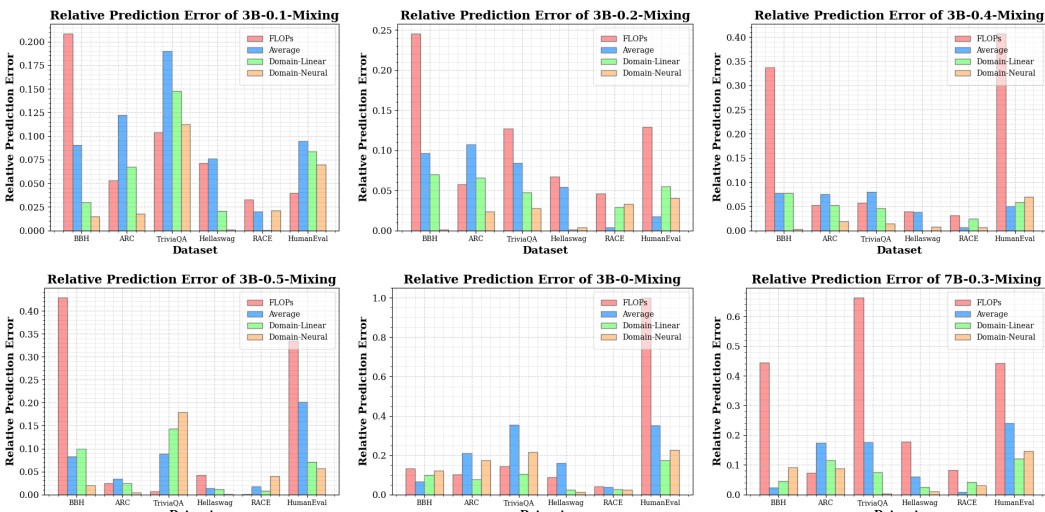

Figure 8: The relative prediction error of various approaches to estimate the Loss-to-Performance curve. Neural network estimation with domain-specific loss as input achieves the best prediction.

pre-training loss of target LLMs, rather than the predictive loss used in §5. We consider the following candidates with different inputs:

(1) **FLOPs:** We adopt the analytical form used to predict the pre-training loss based on training compute (see Eq. 5), only changing the target metric to the downstream performance.

(2) **Average Loss (Average):** We implement a linear regression model to map the average pre-training loss on the whole validation set to the downstream performance.

(3) **Domain Loss via Linear Combination (Domain-Linear):** We apply a linear regression model to correlate pre-training loss across domains with downstream performance.

(4) **Domain Loss via Neural Network (Ours) (Domain-Neural):** We implement a two-layer neural network to map the pre-training loss across domains to the downstream performance. The network configuration and optimization process are introduced in §5.

The fit curves are shown in Fig. 14 (§E) and the results of relative prediction error are shown in Fig. 8. Consistent with the findings in §4, directly estimating the performance based on expended compute (FLOPs) leads to highly inaccurate predictions (FLOPs *vs.* Loss). Pre-training loss serves as a more reliable metric for performance estimation, and decomposing it into domain-specific loss can further enhance prediction accuracy (Average *vs.* Domain Loss). For the predictive models, using neural network estimation can better leverage the abundant data points produced by `FLP-M`, resulting in better performance compared to the linear regression model (Linear *vs.* Neural Network).

## 8 CONCLUSION

This paper introduces a two-stage `FLP` solution to predict downstream performance in LLMs by leveraging pre-training loss. Encouraged by promising preliminary results, we propose `FLP-M`, a core solution for performance prediction that addresses the practical challenges of integrating pre-training data from diverse sources. The effectiveness of `FLP-M` is validated through extensive experiments.

## LIMITATIONS

Our approach `FLP-M` is generally applicable across various data sources, yet currently, it is demonstrated only in binary cases involving code and text data due to computational constraints. Our specific emphasis on the mixing ratio of code is deliberate, reflecting its practical significance in real-world applications. This limitation marks a key area for future expansion.

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

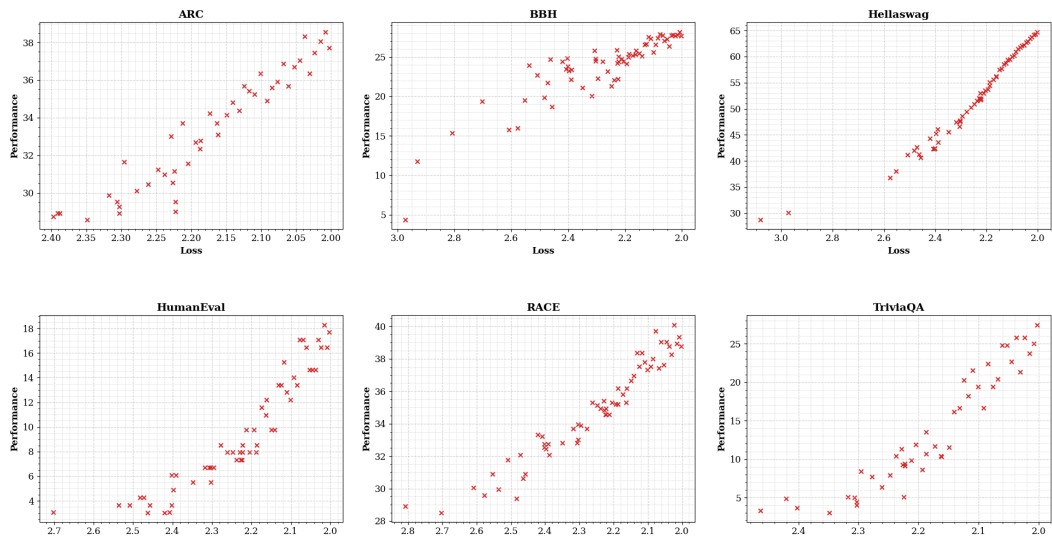

Figure 9: We visualize the relation between pre-training loss and task performance for all LMs that surpass random baseline performance on the target benchmark, observing a generally linear trend.

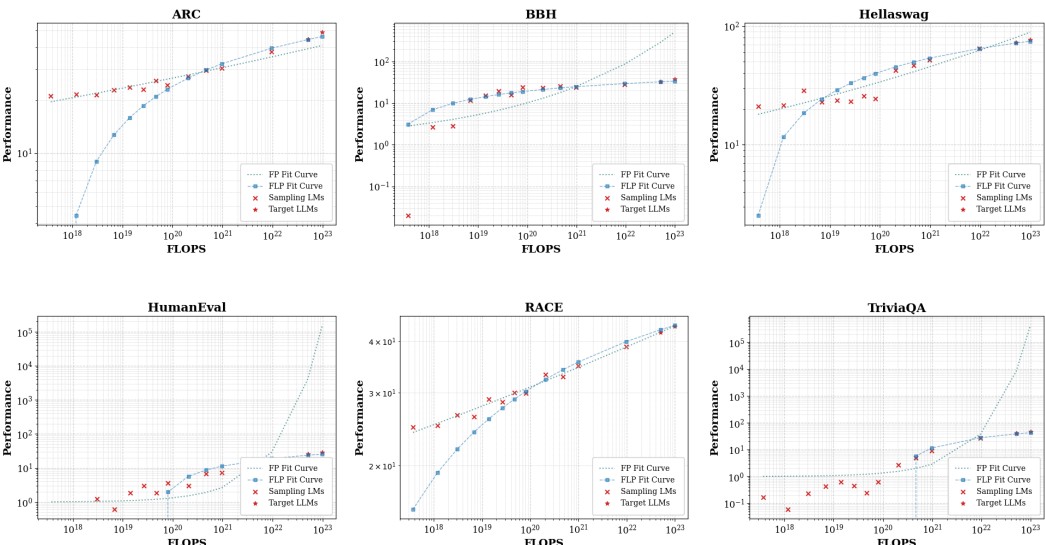

Figure 10: The downstream performance prediction using FP (Achiam et al., 2023) and `FLP` fit curves. `FLP` can better predict the downstream performance of target 7B and 13B LLMs across all evaluation benchmarks, while FP's predictions are very unstable (*e.g.,* HumanEval, TriviaQA).

# APPENDIX

## A  LINEAR RELATION BETWEEN LOSS AND PERFORMANCE

We gather data points from intermediate checkpoints of all sampling LMs and visualize the relationship between pre-training loss and corresponding task performance in Fig. 9. We observe a generally linear trend across all benchmarks, which motivates our selection of linear analytical form to characterize the mapping from pre-training loss to downstream performance.

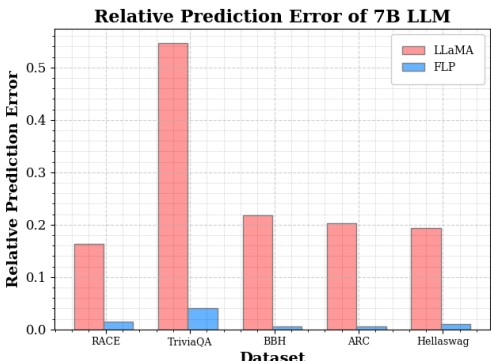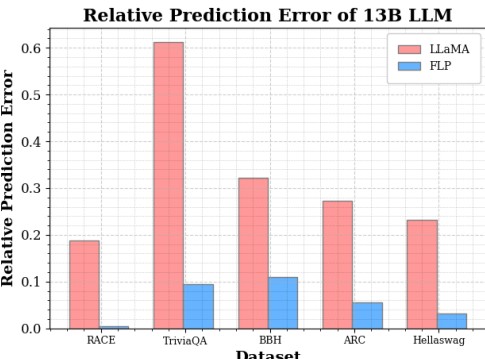

Figure 11: The comparison to the downstream task prediction approach in Llama-3 development (Dubey et al., 2024). We find that initially estimating the negative log-likelihood of the target answer does not effectively predict performance based on our data points.

## B ANALYTICAL FORM TO FIT FLOPS-TO-PERFORMANCE CURVE

We also experiment with the analytical form proposed in Achiam et al. (2023) to estimate the FLOPs-to-Performance curve:

$$\log P(C) = (\frac{C}{C_M})^{\alpha_M}, \tag{6}$$

where $C_M$ and $\alpha_M$ are constant terms to be estimated. The fit curves are shown in Fig. 10. We observe that FLP still consistently outperforms FP across all evaluation benchmarks. In addition, FP can yield very unstable predictions on certain datasets, like HumanEval and TriviaQA, due to a lack of sufficient data for accurate modeling.

## C COMPARE WITH LLAMA-3 APPROACH

We compare with the Llama-3 approach for downstream task prediction (Dubey et al., 2024). They suggest initially estimating the negative log-likelihood (NLL) of the target answer based on the computational cost in FLOPs, followed by using this NLL to model the task performance through a sigmoid function. The comparison results are shown in Fig. 11. We find that the two-stage approach proposed in Dubey et al. (2024) fails to effectively estimate the performance based on our data points, compared to FLP.

## D MMLU EXPERIMENT

Our sampling LMs, up to 3B, exhibit random performance (*i.e.,* 25%) on the MMLU benchmark (Hendrycks et al., 2020). Consequently, these models do not provide effective data points for estimation. Accordingly, we utilize intermediate checkpoints from 7B LLMs to estimate the performance of 13B LLMs on MMLU using FLP. The results are shown in Fig. 12, and the relative prediction error is 3.54%. FLP can also effectively predict the performance on MMLU by leveraging intermediate LMs checkpoints that emerge on this task.

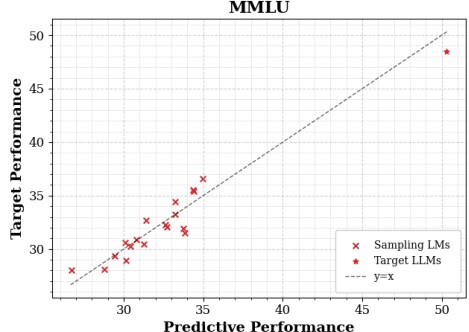

Figure 12: The performance prediction on MMLU using FLP.

## E FLP-M: FIT CURVE FOR ABLATION STUDY

The FLOPs-to-Loss fit curves are in Fig. 13 and the Loss-to-Performance fit curves are in Fig. 14. We observe that $M_4$ in Tab. 3 offers more stable and accurate predictions for domain-specific loss,

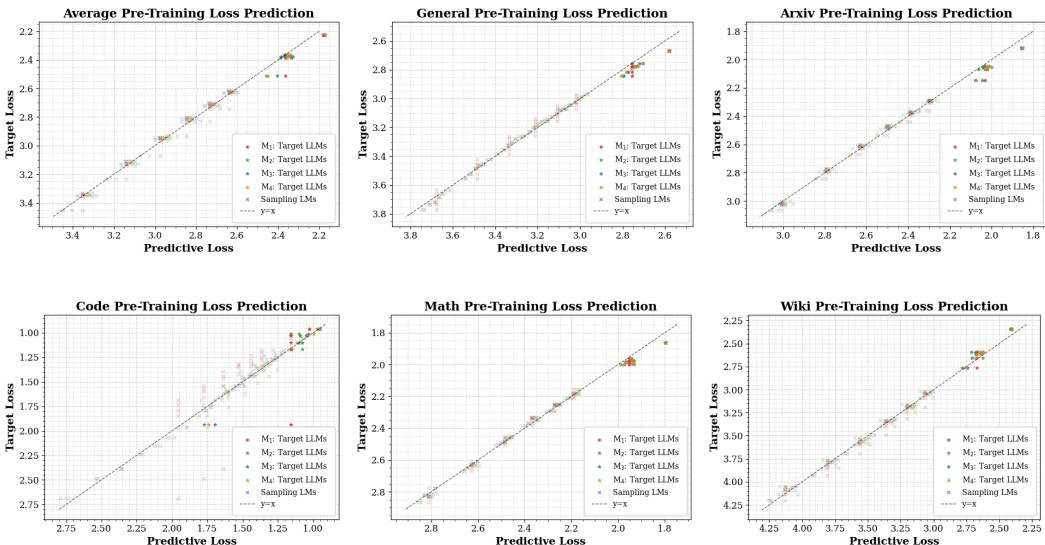

Figure 13: The pre-training loss prediction using various analytical forms. $M_4$ provides more stable and overall more accurate predictions for domain-specific loss.

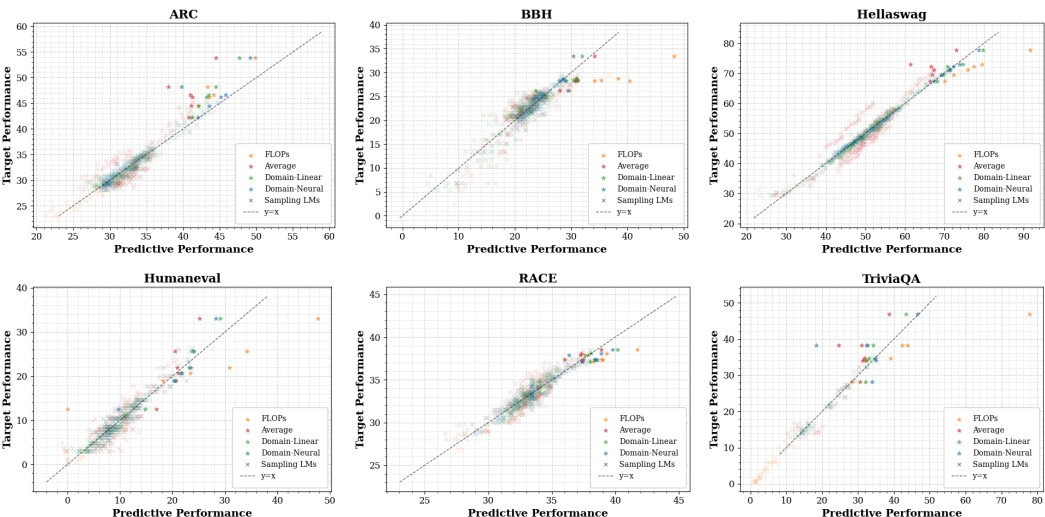

Figure 14: The downstream performance prediction using various approaches. The domain loss coupled with neural network estimation demonstrates the best prediction performance.

with the combined approach of domain loss and neural network estimation delivering the best overall downstream performance prediction.

## F   USING DOMAIN LOSS IN FLP

We explore the application of FLP-M during pre-training on a consistent distribution (the experimental setting described in §4), and compare it with FLP. The fitting curves are shown in Fig. 15 and the results of relative prediction error are shown in Fig. 16. We show that FLP-M fails to effectively predict the performance of target LLMs when sampling LMs are pre-trained on a fixed distribution. This ineffectiveness is attributed to the closely related domain-specific validation losses among the sampling LMs within the same training distribution, which suggests that decomposing the pre-training validation loss yields no additional information in this pre-training setting. Thus, estimating five domain-

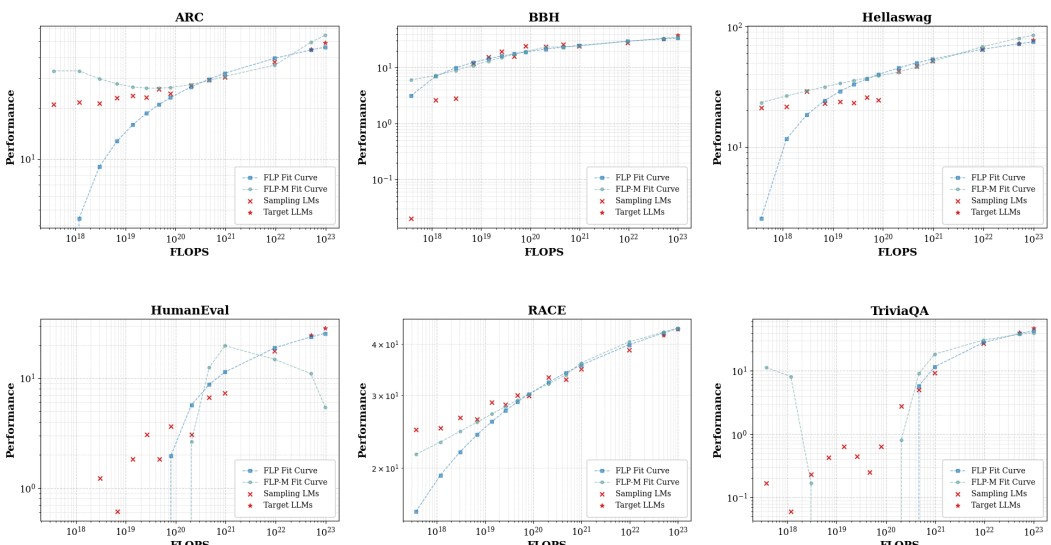

Figure 15: The downstream performance prediction using `FLP` and `FLP-M` fit curves. `FLP` can better predict the downstream performance of target LLMs with 7B and 13B parameters.

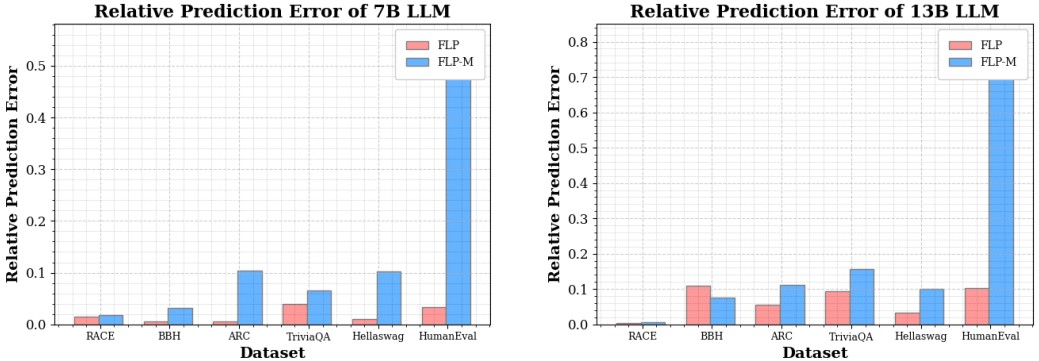

Figure 16: The relative prediction error of 7B and 13B LLMs using `FLP` and `FLP-M`. `FLP` achieves significantly better performance.

specific loss, rather than a single average validation loss, can further increase the risk of error propagation. Moreover, using highly correlated features as neural network inputs may lead to overfitting.

