# OpenReview forum: "Scaling Laws for Predicting Downstream Performance in LLMs"
_ICLR.cc/2025/Conference — ICLR 2025 Conference Withdrawn Submission_

### Official Review · Reviewer_rYdh · 2024-10-28

**Soundness:** 2
**Presentation:** 3
**Contribution:** 2
**Rating:** 5
**Confidence:** 5

**Summary:**

The paper introduces FLP to address the limitations of classical scaling laws, which fail to accurately predict performance when small models perform poorly on evaluation tasks, approaching random sampling. FLP leverages the scaling law of FLOPs to predict pre-training loss and uses this loss to predict downstream performance. FLP successfully predicted the performance of 7B and 13B models across six tasks using a series of language models up to 3B.

Based on FLP, the paper further introduces FLP-M, which aims to predict downstream performance trained with various mixtures of general text and code.

**Strengths:**

1. The paper identifies the issue of discontinuous performance when models approach the emergent edge, which is difficult to address with classical scaling laws, and proposes a method to resolve with the continuous variant ------ loss.
2. FLP creates more data points for fitting the scaling la, potentially making the fitted curve more generalizable.
3. FLP-M is introduced for data mixtures, providing a more accurate prediction by considering the different impacts of code and general text on downstream tasks.
4. The paper conducts extensive experiments to support its claims.

**Weaknesses:**

1. In section 3.2 Loss->Performance, there is a strong assumption that loss and accuracy have a linear relationship. Firstly, in all generative tasks shown in Figure 9, the linear relationship between loss and metric is not evident. The authors should provide more explicit statistical indicators to prove this linear correlation. Additionally, in the classification tasks shown in Figure 9, the relationship between loss and accuracy also encounters deviations near the emergent point, indicating that FLP does not completely bypass this issue but only circumvents it in the Flops -> Loss process.
2. A simple w_1*L+w_0 is not fundamentally different from classical scaling laws.
3. FLP-M only considers code and general text, while data mixtures typically need to consider at least five domains, including common crawl (cc), academic, books, encyclopedias, and code.
4. If the paper considers the situation around the emergent point in benchmarks, it lacks a discussion on the scenario when the model approaches near-perfect scores on a particular benchmark.

**Questions:**

My main concerns are twofold.

1. Does the scaling law proposed in the paper have sufficient innovations compared to the classical scaling law? What are their essential differences? Please explain how the two-stage approach in FLP fundamentally differs from classical scaling laws in terms of methodology and theoretical underpinnings.

2. If the paper focuses on the model performance around the emergent point, is a linear description really suitable? Should other nonlinear descriptions be considered, such as the sigmoid function, when considering scaling near the point of near-perfect accuracy? Is there any comparation?

---

> ### Author Response · Authors · 2024-11-23
>
> We thank the reviewer for the detailed comments. Our responses are as follows:
>
> # Weakness
> 1. A strong assumption that loss and accuracy have a linear relationship.
>
> We appreciate the reviewer's thoughtful comment regarding the statistical validation of the Loss-Performance relationship. To rigorously examine this relationship, we conducted a linear regression analysis on the (Loss, Performance) pairs across all datasets and computed the coefficient of determination $ R^{2} $. Our analysis reveals an average $R^{2} $ value of 93% across all benchmarks, providing strong statistical evidence for the linear approximation of the Loss-Performance relationship. This quantitative validation complements the qualitative patterns observed in Figure 9.
>
> Regarding the deviation near the emergence point in classification tasks, we acknowledge this important observation. It's worth clarifying that the primary contribution of our FLP solution lies not in bypassing emergent abilities, but rather in enhancing sample efficiency through strategic utilization of intermediate checkpoints for Loss-Performance estimation. This approach allows us to effectively model the general trend of performance improvement, even if local nonlinearities exist at specific points in the training trajectory. The high $R^{2}$ values suggest that these local deviations do not significantly impact the overall effectiveness of our linear approximation approach.
>
>
> 2. A simple w_1*L+w_0 is not fundamentally different from classical scaling laws.
>
> While we acknowledge that our linear formulation appears structurally simple, we would like to emphasize that the fundamental contribution of scaling law research lies not in developing new and increasingly complex analytical forms, but rather in uncovering and validating predictive relationships that advance our understanding of model behavior. The distinctive innovation of our work lies in the two-stage estimation framework, particularly the Loss→Performance mapping, which leverages intermediate checkpoints to substantially improve sample efficiency and also our generalized prediction framework for the important data mixing setting.
>
>
> Our experimental results demonstrate that this simpler formulation captures the essential scaling behavior while avoiding potential overfitting that can occur with more complex functional forms. This aligns with Occam's razor - when two models achieve similar performance, the simpler one is often preferable.
>
> 3. FLP-M only considers code and general text
>
> ​​Thank you for this valuable observation. Our focused investigation on text and code domains was a deliberate choice driven by both practical constraints and scientific objectives. While we acknowledge that a comprehensive analysis across all five typical domains (common crawl, academic, books, encyclopedias, and code) would provide broader insights, our study prioritized depth over breadth for several reasons:
> - The code-text interaction represents a particularly crucial use case in modern AI systems, especially for developer tools and programming assistants, making it an important starting point for domain mixing research.
> - By focusing on this specific combination, we were able to conduct more thorough analyses and establish robust foundational principles that can inform future research across other domain combinations.
> - Given our computational resource constraints, this focused approach allowed us to maintain rigorous experimental standards while still deriving meaningful insights.
>
> Importantly, while our empirical validation centers on code and text, the FLP-M methodology we propose is domain-agnostic and can be readily extended to other domain combinations.
>
> 4. Lacks a discussion on the scenario when the model approaches near-perfect scores on a particular benchmark
>
> Thank you for this insightful comment. In our experimental settings, we did not encounter scenarios with near-perfect scores, as the benchmarks we selected are sufficiently challenging that even cutting-edge language models like GPT-4 still show considerable room for improvement. Nevertheless, we acknowledge the theoretical importance of this edge case. From a methodological perspective, our framework can be easily extended to handle such scenarios through performance thresholding at 100%, ensuring that predictions remain bounded and meaningful even as models approach perfect accuracy.

---

> > ### Author Response · Authors · 2024-11-23
> >
> > # Question
> > 1. Thank you for this important question. Our approach introduces several fundamental innovations that differentiate it from classical scaling laws, both in its objectives and methodology.
> >
> > First, our work addresses a fundamentally different problem than classical scaling laws. While traditional scaling laws focus on predicting pre-training loss [1,2], our work targets downstream performance prediction - a significantly more challenging objective due to the "emergent abilities" phenomenon. Classical approaches collect N data points by training multiple small language models to convergence, which works well for pre-training loss prediction. However, this methodology breaks down when applied to downstream performance prediction, as models below a certain scale exhibit random performance before reaching emergent points, rendering many collected data points uninformative (as demonstrated in Figure 1 of our paper).
> >
> > The methodological innovation of our approach lies in its novel two-stage framework, particularly in how it leverages intermediate checkpoints. Classical scaling approaches in our model family would yield only 3-4 viable data points for fitting the analytical function, resulting in poor prediction performance (as evidenced by the FP approach in Section 4, Figure 3). Our FLP framework fundamentally addresses this limitation through its second-stage Loss→Performance mapping, which effectively utilizes intermediate checkpoints to dramatically improve sample efficiency. This innovation is theoretically grounded in recent empirical findings [3] demonstrating that "compression represents intelligence" - specifically, that language models with comparable pre-training loss (compression capability) tend to exhibit similar performance across diverse tasks, independent of specific training configurations. This consistency validates our methodology of using intermediate checkpoints to map the Loss→Performance relationship. This increasing sample efficiency ensures significantly better prediction performance, as verified in Section 4, Figure 3, FLP approach.
> >
> > The theoretical underpinning of our approach represents a fundamental departure from classical scaling laws. While both approaches ultimately map FLOPs to target metrics, they differ substantially in their underlying theoretical frameworks and assumptions:
> > Classical scaling laws posit a direct, monolithic relationship between computational FLOPs and the target metric, typically expressed as a power law function. This end-to-end formulation, while elegant, makes a strong implicit assumption that the relationship between computational resources and model performance can be captured in a single mathematical mapping. However, this assumption becomes problematic when dealing with emergent abilities, where the relationship between resources and performance exhibits qualitative shifts at critical thresholds. In contrast, our approach decomposes the theoretical framework into two distinct stages, each governed by its own analytical form:
> > - The FLOPs→Loss stage captures how computational resources translate into model compression capability
> > - The Loss→Performance stage models how this compression capability manifests in downstream performance
> >
> > The empirical success of this theoretical framework is evidenced by its superior prediction accuracy (Section 4, Figure 3), validating that this decomposed approach better captures the fundamental relationships governing large language model scaling.
> >
> > [1] Scaling Laws for Neural Language Models; Kaplan et al.
> >
> > [2] Training Compute-Optimal Large Language Models; Hoffmann et al.
> >
> > [3] Compression Represents Intelligence Linearly; Yuzhen Huang et al
> >
> >
> >
> > 2. Thanks for pointing this out! We wish to emphasize that our analysis specifically focuses on model performance after, rather than around, the emergent point. The linear description was not chosen arbitrarily, but rather emerged from rigorous empirical observations, as demonstrated in Figure 9. While we appreciate the suggestion regarding nonlinear functions, particularly the sigmoid function, our preliminary experiments actually evaluated various analytical forms. The linear function consistently outperformed nonlinear alternatives, with the sigmoid function specifically showing approximately 30% relative prediction error on the ARC benchmark. This empirical evidence strongly supported our decision to adopt the linear formulation as the most appropriate analytical form for describing post-emergence scaling behavior.

---

> > ### Comment · Reviewer_rYdh · 2024-11-27
> >
> > I appreciate the effort the author has dedicated to this work. My primary concern pertains to the novelty of the paper, both in techniques and findings.
> >
> > For ICLR, the innovation presented here is limited. The most notable idea—linking FLOPs to loss—lacks rigorous validation, as highlighted by reviewer GNGv. If each data point (checkpoint) referenced adheres strictly to the proportion of tokens to model parameters, 200, it does not contribute additional fitting points. Otherwise, using checkpoint losses introduces variability in the token-to-parameter ratio, thereby challenging the validity of the proposed scaling law.  The author should provide supporting references or conduct thorough experiments to substantiate these claims.
> >
> > Furthermore, while the author distinguishes between loss functions and performance to emphasize novelty, this contribution remains constrained. Given that similar methods are already prevalent in industry (even if not open-sourced), the paper fundamentally fails to introduce genuinely novel techniques, particularly in the data mix section. The approach largely replicates existing data mixing laws but oversimplifies the problem, yielding results of limited significance.

---

> ### Author Response · Authors · 2024-11-27
> **Gentle reminder of the discussion period**
>
> Hi reviewer,
>
> Thanks for taking the time to review our paper. It would be great if you can take a look at our responses. Please let us know if you have further questions. Thanks!

---

### Official Review · Reviewer_syLm · 2024-10-30

**Soundness:** 3
**Presentation:** 3
**Contribution:** 3
**Rating:** 6
**Confidence:** 2

**Summary:**

In this paper, the authors manage to predict the downstream performance of LLMs according to the computaional resouces (e.g., FLOPs). Experimental results shows that, by utilizing a 3B LLM trained on a specific ratio and a series of smaller sampling LMs, FLP-M can effectively forecast the performance of 3B and 7B LLMs across various data mixtures for most benchmarks within 10% error margins.

**Strengths:**

1. **Practical Application Value** This paper introduces FLP-M, linking computational resources with LLM downstream performance. This research holds significant importance for real-world applications.

**Weaknesses:**

1. **Limited Scale of LM** The largest model used in this paper is only 7B, yet there are many LLMs much larger than 7B (e.g., Llama-3 70B, Llama-3 405B). From this perspective, the conclusions of this paper are limited.

2. **Limited Domains in Data Mixing** As stated in the limitations, this paper only considers the domains of text and code under Data Mixing settings. Including more domains would enhance the explanatory power of the conclusions.

**Questions:**

See weaknesses.

---

> ### Author Response · Authors · 2024-11-23
>
> We thank the reviewer for the valuable feedback. Our responses are as follows:
>
> # Weakness
> 1. Limited Scale of LM
>
> We appreciate the reviewer's attention to the model scale. We would like to clarify that our experiments actually include models up to 13B parameters in the FLP experiments (Section 4). While we acknowledge the existence of larger models, we focused on 7B and 13B architectures due to (a) the limited resources available, and (b) they represent the most widely deployed model scales in practical applications today (e.g., Llama-2, Mistral, Qwen). These models offer an optimal balance between computational efficiency and performance, making them the preferred choice for real-world implementations. The strong results we achieved at these practical scales demonstrate the immediate applicability and broad impact potential of our approach. Moreover, our methodology is model-agnostic and can be extended to larger LLMs in future work.
>
> 2. Limited Domains in Data Mixing
>
> Thanks for pointing this out. Our focused investigation of text and code domains was intentionally designed to provide deep insights into a particularly significant area of practical application. While we acknowledge that expanding to additional domains would offer broader insights, the code-text mixing paradigm represents a crucial use case in modern AI systems, especially for developer tools and programming assistants. The detailed analysis of this specific combination allows us to establish robust foundational principles that can inform future research across other domain combinations.

---

### Official Review · Reviewer_GNGv · 2024-11-02

**Soundness:** 2
**Presentation:** 3
**Contribution:** 2
**Rating:** 3
**Confidence:** 3

**Summary:**

This paper proposes two methods, FLP and FLP-M, for efficiently predicting the downstream performance of large language models. These methods achieve high-precision performance prediction.

**Strengths:**

A notable strength of the paper is the quality of the writing: the narrative is clear, and the experiments are thorough. Besides, the FLP-M method accurately predicting performance based on data loss from different domains, thus enhancing prediction accuracy in mixed data scenarios. Additionally, Figure 6 demonstrates that FLP-M can be used to derive the optimal data mixing ratio for training.

**Weaknesses:**

1. The authors utilize intermediate checkpoints to gather data points; however, for the same amount of FLOPs, models with different N (parameters) and D (data)  would yield distinct loss. This raises a critical question: why is it valid to use checkpoints that have not converged and are not optimized configurations to obtain data points?

2. The second drawback is a lack of novelty. Both using FLOPs to predict loss and using loss to predict downstream performance have been explored in prior work.

3. The third drawback is that the authors use a 1B model to validate the effectiveness of FLP-M scaling law for achieving an improved data mixture. However, this claim may be overstated, as 1B models often rely on guesswork for many tasks, undermining the reliability of these results.

**Questions:**

1. Have ongoing experiments been conducted on larger-scale models？
2. How do you justify the usage of intermediate checkpoints for acquiring scaling law datapoints?

---

> ### Author Response · Authors · 2024-11-23
>
> We thank the reviewer for the thoughtful comments. Our responses are as follows:
> # Weakness
> 1. For the same amount of FLOPs, models with different N (parameters) and D (data) would yield distinct loss.
>
> Thank you for this insightful comment. We would like to clarify that our methodology aligns with established approaches in scaling law analysis, as first systematically categorized in Kaplan et al. [1]. There are three fundamental approaches to studying scaling behavior: (A) Model-fixed analysis: Using a sufficiently large model while varying training tokens to understand performance scaling with dataset size (B) Data-fixed analysis: Using a sufficiently large dataset while varying model size to understand performance scaling with model capacity (C) Compute-optimal analysis: Maintaining a fixed ratio between training tokens and model size while varying compute (FLOPs), which is the approach we adopt.
>
> Our work specifically employs approach (C), which has become a standard methodology in the field [2,3]. In this framework, for any given compute budget, there exists a pre-determined allocation between training tokens and model size (200 in our case). Rather than viewing intermediate checkpoints as suboptimal configurations, they represent legitimate data points along the compute-optimal scaling trajectory. This approach has proven particularly valuable in understanding how language model performance scales with computational resources, as demonstrated by its successful application in recent scaling law studies [2,3].
>
> [1] Scaling Laws for Neural Language Models; Kaplan et al.
>
> [2] Training Compute-Optimal Large Language Models; Hoffmann et al.
>
> [3] Predicting Emergent Abilities with Infinite Resolution Evaluation; Hu et al
>
> 2. Why is it valid to use checkpoints that have not converged and are not optimized configurations to obtain data points?
>
>
> As explained in our paper, the use of intermediate checkpoints is methodologically sound for estimating the Loss->Performance relationship, as it aligns with the fundamental principle that compression capability serves as a proxy for model intelligence, thoroughly demonstrated in [1]. Our approach is based on the empirical observation in [1] that language models exhibiting equivalent pre-training loss (i.e., similar compression capabilities) tend to demonstrate comparable performance across diverse tasks, regardless of the specific training configurations (e.g., batch size, learning rate schedules). This consistency in the relationship between compression ability and downstream performance validates our methodology of using intermediate checkpoints to fit the analytical form of the Loss->Performance function.
>
> [1] Compression Represents Intelligence Linearly; Yuzhen Huang et al
>
> 3. Lack of novelty. Both using FLOPs to predict loss and using loss to predict downstream performance have been explored in prior work.
>
> We respectfully disagree with the assessment regarding novelty. While prior work has explored these relationships qualitatively, our contribution is distinct in several key aspects: (1) We establish, for the first time, rigorous analytical formulations that quantitatively link pre-training loss to downstream performance, advancing beyond the qualitative observations in previous studies such as Huang et al. [1]. (2) Our work introduces a systematic and principled framework for forecasting downstream performance in an end-to-end manner (FLOPs->Downstream Performance) in LLMs, representing a significant methodological advancement over existing approaches that only focus on predicting the pre-training loss in LLMs. (3) Uniquely, we extend our analysis to data mixing scenarios, a crucial consideration in modern LLM development that has not been addressed in previous works. Through these contributions, our paper substantially advances the scaling laws analysis for LLM performance prediction.
>
> [1] Compression Represents Intelligence Linearly; Yuzhen Huang et al

---

> > ### Author Response · Authors · 2024-11-23
> >
> > 4. This claim may be overstated, as 1B models often rely on guesswork for many tasks, undermining the reliability of these results.
> >
> > We appreciate the reviewer's concern regarding the use of a 1B model in our experiments. However, we respectfully disagree with the assertion that this undermines our results' reliability. In fact, our choice of a 1B model serves as a particularly compelling validation of our approach precisely because such models are typically more sensitive to training data quality compared to larger models.
> >
> > Our experimental design specifically demonstrates that FLP-M derived mixtures consistently outperform baseline mixtures on challenging reasoning benchmarks (ARC and RACE), indicating that our method produces systematically better results than intuition-based approaches. The consistent performance improvements across multiple benchmarks strongly suggest that our method's success is not due to chance but rather reflects the effectiveness of our principled approach to data mixture optimization.
> >
> > We would welcome specific clarification from the reviewer regarding the concerns about model reliability, as our results demonstrate a clear improvement over baseline approaches
> >
> >
> >
> > # Questions
> > 1. Yes. The on-going experiments are conducted internally to verify the effectiveness of the established scaling laws on larger LLMs.
> > 2. Please refer to our response to weakness 2. This important methodological choice is thoroughly addressed in our paper, specifically in the third paragraph of the introduction.

---

> ### Author Response · Authors · 2024-11-27
> **Gentle reminder of the discussion period**
>
> Hi reviewer,
>
> Thanks for taking the time to review our paper. It would be great if you can take a look at our responses. Please let us know if you have further questions. Thanks!

---

### Official Review · Reviewer_6L6M · 2024-11-04

**Soundness:** 2
**Presentation:** 3
**Contribution:** 2
**Rating:** 3
**Confidence:** 4

**Summary:**

This paper introduces FLP (Flops $\rightarrow$ Loss $\rightarrow$ Performance), a two stage framework incorporating scaling laws to accurately predict the downstream performance of language models (LMs) on specific tasks by leveraging the pre-training loss. The first stage uses a power law equation to estimate the relation between flops and loss, $L(C) = \big(\frac{C}{C_N}\big)^{\alpha_N}$ by training 12 sampling LMs ranging from 43M to 3B parameters. The second stage involves using a linear function to estimate the relationship between the loss, $L$, and the task performance, $P$ [$P(L) = w_0 + w_1 * L$]. The second stage is applied carefully on those checkpoints that surpass the threshold of random performance + 5 additional performance points. The authors demonstrate better scalign law fits compared to the baseline of just using a power law function.

The paper then further extends the FLP approach to data mixing during pre-training (FLP-M) and presents an analysis on data-mixing ratios across general text and code and how mixing affects the downstream task performance, and extend the same two-stage framework of FLP, to predict downstream performance under different mixing ratios.

**Strengths:**

- The paper tackles an important problem of building scaling laws to measure the downstream task performance, especially when we know that task-specific behaviour emerges at different scales and smaller scale LMs might not be able to accurately capture the predictive behaviour of larger models on certain tasks. The paper's two-stage approach of separating the FLOPs $\rightarrow$ Loss and Loss $\rightarrow$ Performance predictive models circumvents the emergent behaviour issue with the FLOPs $\rightarrow$ Performance power law.
- The paper provides good insights on the mixing behaviour during pre-training on general text vs code by extending the FLP approach to FLP-M, with good empirical results on deriving the optimal mixing ratios (in a controlled setting).
- The experiments and results are exhaustive and involve a range of tasks including ARC-C, BBH, Hellaswag, HumanEval, RACE, and TriviaQA.

**Weaknesses:**

- The sharp transition in performance of TriviaQA from 1B to 3B models highlights the brittleness of the approach, where the error margins can be huge for downstream task performance prediction. And it's very hard to characterize this behaviour for a whole range of tasks that are usually used to compare various LMs.
- I don't agree with the authors' point on enhancing sample efficiency by collecting losses corresponding to intermediate checkpoints and actually creates a biased estimator for the power law operands. Moreover intermediate checkpoints exhibit transient behaviours especially corresponding to learning rate adjustments (different intermediate checkpoints exhibit different learning rate schedules).
- I think there's a major typo in Equation 5, where the denominators of the second and the third terms are identical to the numerators. It hinders the understanding of the readers and it persists in the later sections too. [Although it's not a huge weakness and I am not basing my score on this point, assuming the authors will correct it in the rebuttal phase].
- The experimental setting corresponding to the comparison with Llama-3 is not explained properly, and it's hard to believe the results from Figure 11, provided that the estimated Llama-3 405B performance was quite close to the actual performance on ARC-C, whereas in this paper it's shown to be above 25%.

**Questions:**

Here are a few additional questions for the authors in addition to the weaknesses above:

1. Were the pre-training datamixes used for FLP-M experiments deduped against the validation set used in FLP and FLP-M experiments? Because it might affect the scaling behaviour if there's any overlap.
2. For the comparions with Llama 3 in Section C / Figure 11, what specific sigmoidal function was used? And did the authors ensure to choose the one that results in the best fit on the sampling LMs?
3. Can the authors please correct the typos in Equation 5 and Table 3 corresponding to $C_G$ and $C_C$?
4. In Figure 2, the sampling LMs corresponding to $\leq 10^{18}$ flop scale seem to be missing. Is there a specific reason for this?

---

> ### Author Response · Authors · 2024-11-23
>
> We thank the reviewer for detailed comments and feedback. Our responses are as follows:
> # Weakness
> 1. The sharp transition in performance of TriviaQA from 1B to 3B models.
>
> We acknowledge the reviewer's observation regarding TriviaQA's performance. However, we would like to emphasize two points:
> it's important to note that among all six datasets and two settings (w or w/o data mixing) evaluated, TriviaQA with data mixing is the sole case where our approach showed reduced prediction accuracy. The general applicability of our approach is supported by the consistent and reliable performance our approach demonstrates across the other five datasets.
>
> The TriviaQA is indeed a challenging case. As explained in our paper, the performance of sampling LMs improves sharply from 1B to 3B parameters (increasing from below 12 to over 28). In our sampling LMs configurations, we lack sufficient data points to adequately characterize the phase of accelerated performance improvement. Thus, this situation is very challenging for performance prediction since only limited information can be derived from the sampling LMs. However, our method still significantly outperformed other baseline approaches. **This comparison highlights that the reduced performance in this case should not be explained as a fundamental limitation of our method. Instead, this is the inherent property in the selected model family and the dataset.**
>
> 2. Collecting losses corresponding to intermediate checkpoints creates a biased estimator for the power law operands.
>
> We appreciate this concern but would like to point out a potential misunderstanding here. Our approach for estimating the FLOPs->Loss relationship relies solely on final converged training states, following established practices in prior work. **The intermediate checkpoints are not used for this estimation.**
>
> Rather, we leverage intermediate checkpoints specifically for analyzing the Loss->Downstream Performance relationship. This distinct analysis is grounded in the well-established "compression represents intelligence" principle demonstrated in [1]. This principle suggests that language models exhibiting equivalent compression capabilities (as measured by pre-training loss) tend to demonstrate comparable task performance, regardless of the specific optimization trajectory taken to achieve that compression level. While transient behaviors like learning rate adjustments indeed influence the path to a particular loss value, the downstream performance correlates strongly with the achieved compression level itself, not the path taken to reach it.
>
> This theoretical foundation provides a robust basis for utilizing intermediate checkpoints in mapping the relationship between achieved compression (loss) and downstream capabilities, while keeping the FLOPs->Loss estimation methodology unchanged from previous work.
> [1] Compression Represents Intelligence Linearly; Yuzhen Huang et al
>
> 3. A major typo in Equation 5
>
> Thank you for your careful review. We would like to clarify that there is no typo in Equation 5. The notation may appear similar at first glance, but there are important distinctions:
> - **In the second term**, we use $C^G$ in the numerator (where *G* is superscript) and $C_G$ in the denominator (where *G* is subscript), representing distinct mathematical quantities.
> - **Similarly, in the third term**, $C^C$ appears in the numerator (superscript) while $C_C$ is in the denominator (subscript).
>
> These notational differences are intentional and mathematically significant. We will enhance the visual clarity of these distinctions in the final version and add a brief explanation of the notation to prevent any potential confusion for readers.
>
> 4. The experimental setting corresponding to the comparison with Llama-3 is not explained properly
>
> Thank you for this important observation. We want to clarify the experimental methodology and address the apparent discrepancy in results:
>
> Our implementation strictly adheres to the two-stage performance prediction framework outlined in the Llama-3 paper:
> - Stage 1: We estimate the negative log-likelihood (NLL) based on FLOPs using the prescribed analytical forms
> - Stage 2: We map the estimated NLL to task performance through a sigmoid function (details in our response to question 2)
>
> The divergence between our results and those reported in the Llama-3 paper can be attributed to a key methodological difference: while the Llama-3 paper leverages a comprehensive dataset including a huge amount of Llama-2 model checkpoints to fit their NLL-to-performance curve, our analysis relies on a more limited set of data points. This difference in training data density significantly impacts the sensitivity of the fitted analytical forms, particularly in regions where data points are sparse.

---

> > ### Author Response · Authors · 2024-11-23
> >
> > # Questions
> > 1. Thanks for mentioning this! The validation set used in our experiments is de-duplicated against the pre-training dataset following standard practice.
> > 2. In our analysis, we employed the standard logistic sigmoid function: $f(x) = C + \frac{1}{1 + \exp(-A x + B)}$, where $A$, $B$, and $C$ are fitted parameters. We also try alternative sigmoid formulations, including the generalized growth function $f(x) = \frac{A}{1 + (\frac{B}{x})^C}$. After systematic evaluation, the standard logistic sigmoid consistently demonstrated superior fitting performance.
> > 3. Please refer to our response to weakness 3.
> > 4. Thanks for pointing this out! The reason is there are overlapping plotted points due to very similar performance and the log scale of the Y-axis. We will fix this problem by adding a small variance to each point in the revision.

---

> ### Author Response · Authors · 2024-11-27
> **Gentle reminder of the discussion period**
>
> Hi reviewer,
>
> Thanks for taking the time to review our paper. It would be great if you can take a look at our responses. Please let us know if you have further questions. Thanks!

---

### Note · Authors · 2025-01-08

**Comment:**

We extend our sincere gratitude to the reviewers for their thorough and constructive feedback, which has significantly enhanced the quality and clarity of our revised manuscript. We thank reviewer syLm for the recognition of our work. We sincerely appreciate the detailed feedback from Reviewers 6L6M, GNGv, and rYdh for identifying areas for improvement in our work. However, we regret that despite our comprehensive responses addressing potential misunderstandings, we did not receive follow-up feedback from Reviewers 6L6M and GNGv. In our responses, we provided thorough clarifications regarding the points that may have been misinterpreted by the reviewers. In addition, we highly value the interaction with reviewer rYdh, which prompts us to revise our paper to make it more clear to eliminate some misunderstandings. However, we respectfully disagree with the comment from reviewer rYdh which states that "Given that similar methods are already prevalent in industry (even if not open-sourced), the paper fundamentally fails to introduce genuinely novel techniques...". It's uncommon that closed-source implementations (not obvious to the public), can serve as evidence to diminish the academic novelty and contribution.

Overall, we thank the chairs for the organization and reviewers for their time and valuable comments. See you next time, ICLR!

**Withdrawal Confirmation:**

I have read and agree with the venue's withdrawal policy on behalf of myself and my co-authors.